# Natural Killer Cell Responses during Human γ-Herpesvirus Infections

**DOI:** 10.3390/vaccines9060655

**Published:** 2021-06-15

**Authors:** Christian Münz

**Affiliations:** Viral Immunobiology, Institute of Experimental Immunology, University of Zürich, 8057 Zürich, Switzerland; christian.muenz@uzh.ch; Tel.: +41-44-635-3716

**Keywords:** Epstein–Barr virus (EBV), Kaposi sarcoma associated herpesvirus (KSHV), NKG2D, NKG2A, DNAM-1

## Abstract

Herpesviruses are main sculptors of natural killer (NK) cell repertoires. While the β-herpesvirus human cytomegalovirus (CMV) drives the accumulation of adaptive NKG2C-positive NK cells, the human γ-herpesvirus Epstein–Barr virus (EBV) expands early differentiated NKG2A-positive NK cells. While adaptive NK cells support adaptive immunity by antibody-dependent cellular cytotoxicity, NKG2A-positive NK cells seem to preferentially target lytic EBV replicating B cells. The importance of this restriction of EBV replication during γ-herpesvirus pathogenesis will be discussed. Furthermore, the modification of EBV-driven NK cell expansion by coinfections, including by the other human γ-herpesvirus Kaposi sarcoma-associated herpesvirus (KSHV), will be summarized.

## 1. Human γ-Herpesvirus Infections and Associated Pathologies

The two human γ-herpesviruses Epstein–Barr virus (EBV) and Kaposi sarcoma-associated herpesvirus (KSHV) are associated with lymphomas and carcinomas [1,2]. They were discovered as human herpesvirus 4 and 8 in 1964 and 1994 in Burkitt’s lymphoma and Kaposi sarcoma, respectively [3,4,5]. Accordingly, these two oncogenic viruses encode proteins that either transform cells on their own or assist cellular oncogenes [6,7,8]. Despite this growth transforming capacity, both viruses are asymptomatically carried by more than 95% of the human adult population in the case of EBV and by more than 50% in Sub-Saharan Africa for KSHV [2,9]. B cells are the main host cells of both viruses. EBV and KSHV colonize these lymphocytes after transmission via saliva exchange. This process is better understood for EBV than for KSHV. B cell infection by EBV in submucosal secondary lymphoid tissues like the tonsils leads to the expression of six nuclear antigens (EBNA1, 2, 3A, 3B, 3C, -LP) and two latent membrane proteins (LMP1 and 2), as well as two Epstein–Barr virus-encoded small RNAs (EBERs) and more than 40 miRNAs in naïve B cells [10,11]. This so-called latency III program is thought to drive B cell differentiation into the germinal center reaction. The resulting centroblasts and -cytes only express EBNA1 and the two LMPs, as well as nontranslated RNAs [11]. This expression program is called latency II and is thought to help the infected B cell to survive the germinal center reaction. This allows EBV-infected B cells to enter the memory B cell pool for long-term persistence [12]. In memory B cells, EBV expresses no viral proteins (latency 0) or transiently EBNA1 during homeostatic proliferation [13]. The expression of EBNA1 and nontranslated viral RNAs is called latency I. Upon B cell activation and plasma cell differentiation from latency 0 or I, EBV switches into lytic replication with infectious virus production [14]. It can then infect polarized epithelial cells from the basolateral side [15,16,17], presumably for an additional round of lytic replication and efficient virus shedding into the saliva. KSHV seems to benefit from this B cell infection by EBV. Efficient KSHV persistence was found to depend on EBV coinfection in mice with reconstituted human immune system compartments [18]. EBV also supported the KSHV infection of human B cells in cell culture [19,20]. In these instances, KSHV infection can be primarily found in EBV-coinfected B cells. Such coinfection can also be observed in primary effusion lymphoma (PEL), a B cell tumor in which KSHV genome maintenance is compromised when EBV is deleted [21,22]. Furthermore, epidemiologically, KSHV infection is associated with EBV infection in African children [23,24]. It remains unclear under which circumstances KSHV infects the endothelial cells from which Kaposi sarcoma originates [2,25]. However, KSHV, similar to EBV, uses the ephrin A2 receptor for epithelial cell infection [16,17,26], presumably prior to shedding into saliva for transmission.

The above outlined evidence suggests that all EBV latency programs that can also be found in EBV-associated malignancies, such as latency I in Burkitt’s lymphoma and gastric carcinoma, latency II in Hodgkin’s lymphoma and nasopharyngeal carcinoma, and latency III in diffuse large B cell lymphoma (DLBCL), are already present in healthy EBV carriers [1,27]. Moreover, EBV-associated B cell lymphomas, PEL and Kaposi sarcoma are increased in patients whose immune system has been weakened by coinfection with the human immunodeficiency virus (HIV) [28]. Therefore, immune control seems to prevent pathology in individuals that are persistently infected with EBV and KSHV. In this review I will discuss the evidence that natural killer (NK) cells are part of this immune control and might be harnessed against EBV- and KSHV-associated pathologies.

## 2. Genetic Evidence for Immune Control of Human γ-Herpesviruses by NK Cells

NK cells seem to play a crucial role in the immune control of herpesvirus infections. Even in the first patient that was described with NK cell deficiency, severe α- (herpes simplex and varicella zoster virus) and β-herpesvirus (cytomegalovirus [CMV]) infections were observed [29]. Later-on mutations in the transcription factor GATA2 that prevent early differentiated CD56^bright^ NK cell development were identified in this and other patients [30]. Interestingly, primary immunodeficiencies that cause NK cell deficiencies, such as the GATA2 mutations, predispose either for CMV or EBV pathologies [31]. Among the NK cell deficiencies that are associated with EBV-positive lymphomas and EBV-driven immunopathologies such as hemophagocytic lymphohistiocytosis (HLH) are mutations in the minichromosome maintenance complex component 4 (MCM4) or interferon regulatory factor 8 (IRF8) [32,33,34]. These deficiencies compromise NK cell maturation to CD56^dim^ cells with increased cytotoxicity. Indeed, primary immunodeficiencies that compromise the cytotoxic machinery of lymphocytes predispose for EBV-associated pathologies [35,36,37]. The respective mutations affect the pore-forming protein perforin or the adaptor proteins Munc13-4 and Munc18-2 that are required for cytotoxic granule fusion with the plasma membrane [38,39,40]. In addition to the cytotoxic machinery itself, receptors that mediate NK cell function are also affected by mutations in some patients with primary immunodeficiencies that predispose for EBV-associated pathologies. These include the activating receptor NKG2D and the activating coreceptors CD2, 2B4 and NTB-A on NK cells [41,42,43,44,45,46,47,48,49,50,51]. In this respect, XMEN (X-linked immunodeficiency with magnesium defect, Epstein–Barr virus (EBV) infection and neoplasia) disease is caused by mutations in the magnesium transporter MagT1 [41,42,43]. This compromises the surface expression of the activating NK cell receptor NKG2D, which is involved in the NK cell recognition of lytically EBV replicating B cells [52]. Furthermore, mutations in the Fc receptor CD16 compromise natural cytotoxicity, but not antibody-dependent cellular cytotoxicity (ADCC) by the affected NK cells [45,53]. In particular, they prevent the efficient recruitment of the activating coreceptor CD2 to the immunological synapse for efficient NK cell killing [45]. Finally, loss-of-function mutations in the SLAM (signalling lymphocytic activation molecule) receptor-associated protein SAP convert costimulatory signals of the two SLAM receptors 2B4 and NTB-A to inhibitory signals, thereby inhibiting lysis of EBV-transformed B cells by NK cells [54,55]. These mutations in patients with primary immunodeficiencies argue for an important role of cytotoxic CD56^dim^CD16^+^CD2^+^NKG2D^+^2B4^+^NTB-A^+^ NK cells in the immune control of EBV.

While an increased predisposition for Kaposi sarcoma was also described for patients with XMEN disease [56], KSHV-specific immune control seems to depend much more on the production of the antiviral cytokine IFN-γ than EBV-specific immune control. Accordingly, mutations in the IFN-γ receptor (IFN-γR1) and STAT4 that elicit IFN-γ transcription after IL-12 signaling increase the risk for Kaposi sarcoma [57,58]. This could indicate that, similar to CMV, early differentiated CD56^bright^ NK cells with a preferential IFN-γ production but less pronounced natural cytotoxicity are more important in KSHV-specific immune control than in EBV-specific immune control.

## 3. NK Cell Phenotype and Function during EBV Infection

In addition to genetic lesions that compromise NK cell differentiation, effector function and stimulation, and thereby affect the immune control of EBV and KSHV infection [37], changes in the magnitude and composition of the NK cell compartment after the respective viral infections point towards their contribution to immune responses to these two oncogenic γ-herpesviruses. This has been primarily investigated during symptomatic primary EBV infection [9,59]. In the United States and Europe, two thirds of the population get infected with EBV before the age of 2, and a substantial proportion of the remaining one third experiences primary infection in the second decade of life [60,61,62]. In 30–50% of cases, delayed primary infections lead to infectious mononucleosis (IM) with a massive expansion of lytic EBV antigen-specific CD8^+^ T cells and the associated immunopathology [63,64]. In addition, NK cells expand during IM [61,65,66,67]. These are in particular CD56^dim^CD16^+/−^ NK cells that are negative for the activating NKG2C receptor and express the inhibitory NKG2A receptor, both binding the nonclassical major histocompatibility complex (MHC) class I molecule HLA-E [66,68] (Figure 1). This NK cell differentiation stage is also still mostly negative for the expression of inhibitory killer immunoglobulin-like receptors (KIRs) that can distinguish haplotypes of classical MHC class I molecules. In good agreement with the phenotypic characteristics that can be deduced from the genetic predispositions for EBV pathologies, this phenotype represents an intermediate differentiation stage after CD56^bright^CD16^−^ NK cells, but prior to CD56^dim^CD16^+^KIR^+^NKG2C^+^NKG2A^−^ adaptive NK cells and terminally differentiated CD56^−^CD16^+^ NK cells [69,70,71] (Figure 1). In contrast to EBV, CMV seems to expand adaptive NKG2C^+^ NK cells [72,73] and even drive KIR^+^ NK cell accumulation in individuals with a genetic deletion of NKG2C [74,75,76] (Figure 1). NKG2A^+^KIR^−^ NK cells that expand during IM preferentially target B cells that undergo lytic EBV replication [52,66,77,78]. In particular, the activating NK cell receptor NKG2D and the activating coreceptor DNAM-1 are involved in the recognition of lytically EBV replicating B cells by NK cells [52]. EBV compromises this recognition in part by the viral miRNA-mediated downregulation of the NKG2D ligand MICB [79]. Nevertheless, NK cells restrict primarily wild-type EBV infection but not lytic replication-deficient BZLF1 knock-out virus infection in mice with reconstituted human immune system components (humanized mice) [78]. Within three months after neonatal human CD34^+^ hematopoietic progenitor cell transfer, this preclinical model of the human immune system primarily reconstitutes CD56^bright^CD16^−^ NK cells, unless IL-15 is supplied either by transgenic expression or improved myeloid cell reconstitution [80,81,82]. NK cells expand after EBV infection in this model and accumulate as CD56^dim^CD16^+^NKG2A^+^KIR^−^ NK cells [78]. Upon NK cell depletion, EBV loads do not only increase, but so does the incidence of monoclonal B cell lymphomas as well as signs of lytic EBV replication in tissue sections [78]. In the absence of NK cell-mediated immune control of lytic EBV replication, CD8^+^ T cells expand dramatically after EBV infection of humanized mice [78]. This massive expansion and immune pathology, such as weight loss, is associated with an elevated proinflammatory cytokine production, reminiscent of IM. Indeed, the protective CD56^dim^CD16^+^NKG2A^+^KIR^−^ NK cell population decreases in frequency during the first decade of a child’s life [66,83] and could thus predispose for IM upon delayed primary EBV infection in the second decade of life. These studies suggest that NKG2A^+^KIR^−^ NK cells preferentially respond to EBV infection and restrict lytic EBV replication which might otherwise drive IM.

## 4. Modulation of NK Cell Responses by KSHV

In certain EBV-associated disease settings and upon coinfections, the composition of the NK cell compartment can change and thereby presumably fail to mediate EBV-specific immune control. Along these lines, it was noted that children with endemic Burkitt’s lymphoma accumulated CD56^−^CD16^+^ NK cells [84]. These are thought to be a further differentiation stage of CD56^dim^CD16^+^ NK cells and share their transcriptional profile to a large extent [85]. Since coinfection with the malaria parasite *Plasmodium falciparum* (Pf) is associated with an increased risk of developing Burkitt’s lymphoma [86] and since certain KIR receptor-HLA ligand pairs are associated with the immune control of this parasite [87,88], coinfection with Pf might be in part responsible for the observed accumulation of CD56^−^CD16^+^ NK cells. This could also occur via the Pf-mediated activation of another coinfection, such as KSHV infection. Indeed, antimalarial parasite antibodies were found to be associated with KSHV seropositivity [89], and KSHV-seropositive African children presented with a higher frequency of CD56^−^CD16^+^ NK cells [90]. Similarly transplant patients with EBV-associated lymphoproliferative disease (PTLD) present with diminished CD56^dim^CD16^+^NKG2A^+^KIR^−^ NK cells, despite similarly high EBV loads as those observed in IM patients [67]. In CMV-positive patients, CD56^dim^CD16^+^NKG2C^+^NKG2A^−^KIR^+^ NK cells accumulate at the expense of this earlier NK cell differentiation stage. Possibly, CMV reactivation after immune suppressive treatment following transplantation drives this NK cell differentiation, away from the NK cell phenotype that protects against lytic EBV replication.

Such NK cell differentiation could also be observed upon EBV-plus-KSHV coinfection of humanized mice [90]. CD56^−^CD16^+^ NK cells accumulate upon KSHV coinfection at the expense of CD56^bright^CD16^−^ NK cells (Figure 1). KSHV, but not EBV viral loads correlate with this accumulation. The cytokines IL-15, IL-18 and IL-27 that are increased during EBV-plus-KSHV coinfection might be at least partially responsible for the observed NK cell differentiation. As previously described in African children with endemic Burkitt’s lymphoma, these CD56^−^CD16^+^ NK cells share many similarities with CD56^dim^CD16^+^ NK cells [85,90]. However, they are even further enriched in the expression of the cytotoxic machinery, containing perforin and granzyme B [90]. KSHV coinfection also drives the expression of CXCR6 and CD69, which are often associated with tissue residency. Despite the high expression of perforin and granzyme B in the CD56^−^CD16^+^ NK cell population, they exercise less natural and antibody-dependent cytotoxicity. Furthermore, they produce less IFN-γ and do not proliferate. Instead, their expression of the ATPase CD39 might rather point to an immune suppressive function [90,91]. Furthermore, at least in Burkitt’s lymphoma patients, CD56^−^CD16^+^ NK cells seem to retain MIP1β/CCL4 production, which can attract CCR5^+^ regulatory T and suppressive myeloid cell populations [92,93]. Patients with Kaposi sarcoma also present with a diminished NK cell cytotoxicity [94,95]. Thus, KSHV coinfection drives the expansion of CD56^−^CD16^+^CXCR6^+^CD39^+^ NK cells, which might suppress immune responses in tissues by ATP hydrolysis and suppressive leucocyte attraction via MIP1β/CCL4 to avoid immune pathology.

In addition, KSHV also employs the direct modulation of NK cell recognition by inhibition of ligand upregulation for the activation of NK cell receptors, of migration of NK cells and by exploitation of certain KIR haplotypes. Along these lines, KSHV downregulates ligands for activating NK cell receptors, such as NKG2D, NKp44, NKp80 and DNAM-1 [96,97,98,99]. The viral ubiquitin ligase K5 downregulates the NKG2D ligands MICA and MICB, the NKp80 ligand AICL, and the DNAM-1 ligands Nectin-2 and CD155 [96,98]. Furthermore, viral ORF54 downregulates NKp44 ligands of an unknown identity independently of ORF54′s UTPase activity [97]. MICB and NKG2D ligands inducing AID upregulation are also targeted by KSHV-encoded miRNA [79,100]. In addition to inhibiting the NK cell recognition of infected cells, the KSHV-encoded viral MIP-II blocks NK cell migration [101]. Finally, Kaposi sarcoma is enriched in patients that encode the activating KIRs KIR3DS1 and KIR2DS1 with their MHC class I ligands [102,103]. How these receptors support Kaposi sarcoma development, however, remains unclear. Nevertheless, these studies suggest that KSHV inhibits NK cell responses by both driving NK cell differentiation to diminish antiviral effector functions and by compromising the NK cell recognition of KSHV-infected cells.

## 5. Harnessing NK Cells against γ-Herpesvirus-Associated Pathologies

These previous studies indicate that some coinfections, possibly including malaria, CMV and KSHV, differentiate NK cells and thereby diminish the immune control of lytic EBV infection by NKG2A^+^KIR^−^ NK cells. Indeed, the control of lytic EBV infection seems important to prevent virus-associated pathologies; not only IM, but abortive early lytic replication also seems required, at least for some EBV-associated malignancies [10,18,104,105,106,107,108,109]. Furthermore, some primary immunodeficiencies that predispose for EBV-associated lymphomas seem to also preferentially affect the T cell-mediated immune control of lytic EBV infection [110]. Thus, the inherent reactivity of NK cells against lytic EBV replication could be harnessed against EBV-associated pathologies.

Along these lines, KIR mismatching could recruit further differentiated NK cell subpopulations, in addition to NKG2A^+^KIR^−^ NK cells, to the immune response against EBV-associated lymphomas [111,112,113]. In humanized mice, KIR ligand (HLA-C1, -C2 and -Bw4) mismatched mixed human immune compartment reconstitution generates an environment for EBV infection, in which some EBV-infected B cells cannot efficiently engage inhibitory KIRs on NK cells [114]. Despite diminished NK cell licensing in trans, this leads to an improved immune control of inflammatory IM like EBV infection. Antibody-mediated NK cell depletion abolishes this improved immune control of mixed hematopoietic lineage reconstitution [114]. These findings suggest that KIR mismatched NK cells can target EBV-infected B cells, possibly via the engagement of the activating receptors NKG2D and DNAM-1 and in the absence of inhibitory KIR signaling.

The recognition of EBV-infected B cells by NK cells beyond lytic replication can be strengthened by providing additional activating receptors to these innate killers. Along these lines, chimeric antigen receptors (CARs) have been explored, which target surface molecules of B cells via a variable antibody fragment linked to T cell receptor- and costimulatory receptor-signaling domains [115,116,117,118]. Initial preclinical and clinical studies have focused on CD19-targeting CARs. In a phase I clinical trial, four of six patients with lymphoma experience completed remission without any signs of cytokine release syndrome (CRS) which is often common to CAR therapies [118]. Beyond CD19, CD22 and CD30 are now also explored for similar treatments [115,119]. Therefore, KIR mismatching might allow one to diminish inhibitory signaling, and CAR expression could enable NK cell-mediated targeting of EBV-infected B cells beyond lytic replication, in order to harness NK cell responses against EBV-associated malignancies.

In contrast to EBV and associated lymphomas, NK cells have not yet been harnessed for clinical use against KSHV-associated malignancies. However, NK cells have been described as killing KSHV-infected cells [120,121,122,123,124,125]. KSHV-mediated downregulation of MHC class I renders infected cells susceptible to NK cell recognition, but virus-induced ICAM-1 and B7-2 (CD86) suppression limits this recognition of PEL cells [124,125]. IFN-γ-mediated upregulation of these molecules might partially restore this recognition [126]. Therefore, IFN-γ-producing cytotoxic NK cells might prove useful for the therapeutic targeting of KSHV-associated malignancies, including PEL.

## 6. Conclusions and Outlook

Studying the human γ-herpesviruses EBV and KSHV reveals that they drive NK cells into distinct differentiation stages. While EBV expands early-differentiated CD56^dim^CD16^+^NKG2A^+^KIR^−^ NK cells, KSHV coinfection leads to the accumulation of terminally differentiated CD56^−^CD16^+^NKG2A^−^KIR^+^CD39^+^CXCR6^+^ NK cells [66,90]. The mechanisms of this differential NK cell repertoire shaping should be further investigated in the future. They can teach us how to expand optimal NK cell populations for therapeutic applications, as is currently proposed for CMV-driven adaptive NK cell expansions [76,127]. They can then be further improved by transgenically expressing additional activations and deletions of inhibitory receptors to target malignancies that are caused by human γ-herpesviruses and tumors in general.

## Figures and Tables

**Figure 1 vaccines-09-00655-f001:**
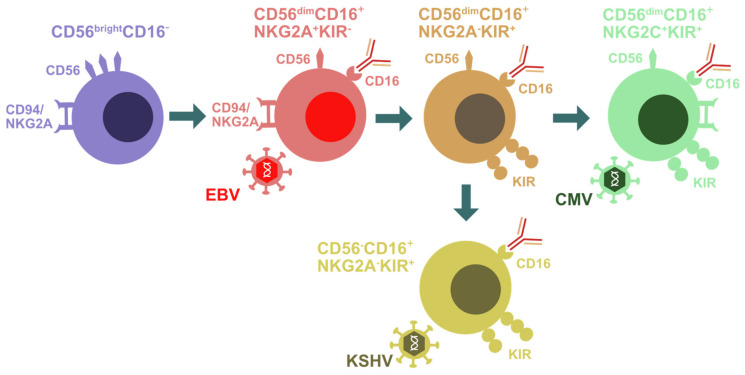
Natural killer (NK) cell differentiation during herpesvirus infection. Early differentiated CD56^bright^CD16^−^ NK cells are primarily found in secondary lymphoid tissues. They give rise to CD56^dim^CD16^+^NKG2A^+^KIR^−^ NK cells that are preferentially expanded during EBV infection. Upon further differentiation, these cells acquire KIRs and lose NKGA. CD56^dim^CD16^+^NKG2A^−^KIR^+^ NK cells seem to then give rise to either CD56^dim^CD16^+^NKG2C^+^KIR^+^ adaptive NK cells that accumulate during CMV infection or to CD56^−^CD16^+^NKG2A^−^KIR^+^ terminally differentiated NK cells that are enriched in KSHV-coinfected individuals.

## Data Availability

Data sharing not applicable. No new data were created or analyzed in this study. Data sharing is not applicable to this article.

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
