# Peer review of "Natural Killer Cell Responses during Human γ-Herpesvirus Infections"

_vaccines, 2021, doi:10.3390/vaccines9060655_

Round 1
Reviewer 1 Report
This is a good review to describe the phenotype, function and genetic changes for NK cells during EBV, KSHV infection. Detailed and systematic study results on NK cells provided solid evidences. The whole paper is generally well written and structured. In my opinion, the manuscript is suitable for publishing for the journal, vaccines, if the authors have more citation papers associated with different types of lymphomas and sarcomas.
more specific:
This is a good review to describe the phenotype, function and genetic changes for NK cells during EBV, KSHV infection. Detailed and systematic study results on NK cells provided solid evidences. The whole paper is generally well written and structured and will benefit the scientists working on the NK cells associated with EBV, KSHV and some types of cancers. In my opinion, the manuscript is suitable for publishing for the journal, vaccines.
Minor revision
The authors should have more citation papers associated with different types of lymphomas and sarcomas since the EBV infection is associated with many kinds of cancers, such as burkitt’s lymphomas, Hodgkin’s lymphomas, nasopharyngeal cancers of epithelial origin, and some types of gastric carcinomas.
Author Response
I thank this reviewer for his/her constructive comments which I have incorporated into the revised manuscript version. The changes are high-lighted below and by underlining in the revised manuscript text.
Reviewer #1
This is a good review to describe the phenotype, function and genetic changes for NK cells during EBV, KSHV infection. Detailed and systematic study results on NK cells provided solid evidences. The whole paper is generally well written and structured. In my opinion, the manuscript is suitable for publishing for the journal, vaccines, if the authors have more citation papers associated with different types of lymphomas and sarcomas.
Additional references for the different EBV and KSHV associated malignancies have now been cited on page 2 of the revised manuscript.
more specific:
This is a good review to describe the phenotype, function and genetic changes for NK cells during EBV, KSHV infection. Detailed and systematic study results on NK cells provided solid evidences. The whole paper is generally well written and structured and will benefit the scientists working on the NK cells associated with EBV, KSHV and some types of cancers. In my opinion, the manuscript is suitable for publishing for the journal, vaccines.
I thank this reviewer for his/her support.
Minor revision
The authors should have more citation papers associated with different types of lymphomas and sarcomas since the EBV infection is associated with many kinds of cancers, such as burkitt’s lymphomas, Hodgkin’s lymphomas, nasopharyngeal cancers of epithelial origin, and some types of gastric carcinomas.
More references for both EBV and KSHV associated malignancies have now been cited on page 2 of the revised manuscript.
Reviewer 2 Report
This is a review of a very important topic by a world expert in the field of NK responses to gamma-herspesvirus. It is a review at a very high level that would benefit scientists with expertise working in these fields of virology and immunology. Much of the evidence reviewed is biased towards the NK response to EBV and its modulations with co-infections such as KSHV. I think the review will benefit from more specific discussions on the response to KSHV and a section discussing the excellent published evidence on strategies deployed by KSHV to evade the NK response. As a guide, I am suggesting publications that should be referenced to make this review more KSHV-comprehensive.
1: Gabaev I, Williamson JC, Crozier TWM, Schulz TF, Lehner PJ. Quantitative Proteomics Analysis of Lytic KSHV Infection in Human Endothelial Cells Reveals Targets of Viral Immune Modulation. Cell Rep. 2020 Oct 13;33(2):108249. doi: 10.1016/j.celrep.2020.108249. PMID: 33053346; PMCID: PMC7567700.
2: Goedert JJ, Martin MP, Vitale F, Lauria C, Whitby D, Qi Y, Gao X, Carrington M. Risk of Classic Kaposi Sarcoma With Combinations of Killer Immunoglobulin- Like Receptor and Human Leukocyte Antigen Loci: A Population-Based Case-control Study. J Infect Dis. 2016 Feb 1;213(3):432-8. doi: 10.1093/infdis/jiv413. Epub 2015 Aug 12. PMID: 26268853; PMCID: PMC4719589.
3: Bekerman E, Jeon D, Ardolino M, Coscoy L. A role for host activation-induced cytidine Yamindeaminase in innate immune defense against KSHV. PLoS Pathog. 2013;9(11):e1003748. doi: 10.1371/journal.ppat.1003748. Epub 2013 Nov 7. PMID: 24244169; PMCID: PMC3820765.
4: Yamin R, Kaynan NS, Glasner A, Vitenshtein A, Tsukerman P, Bauman Y, Ophir Y, Elias S, Bar-On Y, Gur C, Mandelboim O. The viral KSHV chemokine vMIP-II inhibits the migration of Naive and activated human NK cells by antagonizing two distinct chemokine receptors. PLoS Pathog. 2013 Aug;9(8):e1003568. doi: 10.1371/journal.ppat.1003568. Epub 2013 Aug 15. PMID: 23966863; PMCID: PMC3744409.
5: Madrid AS, Ganem D. Kaposi's sarcoma-associated herpesvirus ORF54/dUTPase downregulates a ligand for the NK activating receptor NKp44. J Virol. 2012 Aug;86(16):8693-704. doi: 10.1128/JVI.00252-12. Epub 2012 Jun 6. PMID: 22674989; PMCID: PMC3421743.
6: Guerini FR, Mancuso R, Agostini S, Agliardi C, Zanzottera M, Hernis A, Tourlaki A, Calvo MG, Bellinvia M, Brambilla L, Clerici M. Activating KIR/HLA complexes in classic Kaposi's Sarcoma. Infect Agent Cancer. 2012 Apr 2;7:9. doi: 10.1186/1750-9378-7-9. PMID: 22469025; PMCID: PMC3379936.
7: Nachmani D, Stern-Ginossar N, Sarid R, Mandelboim O. Diverse herpesvirus microRNAs target the stress-induced immune ligand MICB to escape recognition by natural killer cells. Cell Host Microbe. 2009 Apr 23;5(4):376-85. doi: 10.1016/j.chom.2009.03.003. PMID: 19380116.
8: Thomas M, Boname JM, Field S, Nejentsev S, Salio M, Cerundolo V, Wills M, Lehner PJ. Down-regulation of NKG2D and NKp80 ligands by Kaposi's sarcoma- associated herpesvirus K5 protects against NK cell cytotoxicity. Proc Natl Acad Sci U S A. 2008 Feb 5;105(5):1656-61. doi: 10.1073/pnas.0707883105. Epub 2008 Jan 29. PMID: 18230726; PMCID: PMC2234200.
9: Ishido S, Choi JK, Lee BS, Wang C, DeMaria M, Johnson RP, Cohen GB, Jung JU. Inhibition of natural killer cell-mediated cytotoxicity by Kaposi's sarcoma- associated herpesvirus K5 protein. Immunity. 2000 Sep;13(3):365-74. doi: 10.1016/s1074-7613(00)00036-4. PMID: 11021534
Author Response
I thank this reviewer for his/her constructive comments which I have incorporated into the revised manuscript version. The changes are high-lighted below and by underlining in the revised manuscript text.
Reviewer #2
This is a review of a very important topic by a world expert in the field of NK responses to gamma-herpesvirus. It is a review at a very high level that would benefit scientists with expertise working in these fields of virology and immunology. Much of the evidence reviewed is biased towards the NK response to EBV and its modulations with co-infections such as KSHV. I think the review will benefit from more specific discussions on the response to KSHV and a section discussing the excellent published evidence on strategies deployed by KSHV to evade the NK response. As a guide, I am suggesting publications that should be referenced to make this review more KSHV-comprehensive.
1: Gabaev I, Williamson JC, Crozier TWM, Schulz TF, Lehner PJ. Quantitative Proteomics Analysis of Lytic KSHV Infection in Human Endothelial Cells Reveals Targets of Viral Immune Modulation. Cell Rep. 2020 Oct 13;33(2):108249. doi: 10.1016/j.celrep.2020.108249. PMID: 33053346; PMCID: PMC7567700.
2: Goedert JJ, Martin MP, Vitale F, Lauria C, Whitby D, Qi Y, Gao X, Carrington M. Risk of Classic Kaposi Sarcoma With Combinations of Killer Immunoglobulin- Like Receptor and Human Leukocyte Antigen Loci: A Population-Based Case-control Study. J Infect Dis. 2016 Feb 1;213(3):432-8. doi: 10.1093/infdis/jiv413. Epub 2015 Aug 12. PMID: 26268853; PMCID: PMC4719589.
3: Bekerman E, Jeon D, Ardolino M, Coscoy L. A role for host activation-induced cytidine deaminase in innate immune defense against KSHV. PLoS Pathog. 2013;9(11):e1003748. doi: 10.1371/journal.ppat.1003748. Epub 2013 Nov 7. PMID: 24244169; PMCID: PMC3820765.
4: Yamin R, Kaynan NS, Glasner A, Vitenshtein A, Tsukerman P, Bauman Y, Ophir Y, Elias S, Bar-On Y, Gur C, Mandelboim O. The viral KSHV chemokine vMIP-II inhibits the migration of Naive and activated human NK cells by antagonizing two distinct chemokine receptors. PLoS Pathog. 2013 Aug;9(8):e1003568. doi: 10.1371/journal.ppat.1003568. Epub 2013 Aug 15. PMID: 23966863; PMCID: PMC3744409.
5: Madrid AS, Ganem D. Kaposi's sarcoma-associated herpesvirus ORF54/dUTPase downregulates a ligand for the NK activating receptor NKp44. J Virol. 2012 Aug;86(16):8693-704. doi: 10.1128/JVI.00252-12. Epub 2012 Jun 6. PMID: 22674989; PMCID: PMC3421743.
6: Guerini FR, Mancuso R, Agostini S, Agliardi C, Zanzottera M, Hernis A, Tourlaki A, Calvo MG, Bellinvia M, Brambilla L, Clerici M. Activating KIR/HLA complexes in classic Kaposi's Sarcoma. Infect Agent Cancer. 2012 Apr 2;7:9. doi: 10.1186/1750-9378-7-9. PMID: 22469025; PMCID: PMC3379936.
7: Nachmani D, Stern-Ginossar N, Sarid R, Mandelboim O. Diverse herpesvirus microRNAs target the stress-induced immune ligand MICB to escape recognition by natural killer cells. Cell Host Microbe. 2009 Apr 23;5(4):376-85. doi: 10.1016/j.chom.2009.03.003. PMID: 19380116.
8: Thomas M, Boname JM, Field S, Nejentsev S, Salio M, Cerundolo V, Wills M, Lehner PJ. Down-regulation of NKG2D and NKp80 ligands by Kaposi's sarcoma- associated herpesvirus K5 protects against NK cell cytotoxicity. Proc Natl Acad Sci U S A. 2008 Feb 5;105(5):1656-61. doi: 10.1073/pnas.0707883105. Epub 2008 Jan 29. PMID: 18230726; PMCID: PMC2234200.
9: Ishido S, Choi JK, Lee BS, Wang C, DeMaria M, Johnson RP, Cohen GB, Jung JU. Inhibition of natural killer cell-mediated cytotoxicity by Kaposi's sarcoma- associated herpesvirus K5 protein. Immunity. 2000 Sep;13(3):365-74. doi: 10.1016/s1074-7613(00)00036-4. PMID: 11021534
I thank this reviewer for this suggestions and have now included a discussion of KSHV mediated immune escape from NK cell recognition of infected cells on page 5 of the revised manuscript. The above listed articles were cited in this discussion.
Reviewer 3 Report
In the present paper the author reviews how three different g-Herpesviruses determines a control of the maturation and differentiation of NK cells. In my opinion, the review captures the point proposed in the title and provides a very clear picture of how Herpesviruses are able to modulate the immune response of NK cells.
The strength of this work consists in highlighting how the shaping of NK cells is dependent on the replication phase of Herpesviruses, which until a few years ago was considered a sporadic event and of little interest compared to the latency phase.
The weakness of the work is probably related to the absence of additional figures that could help the reader to promptly acquire information.
I also believe that the remodeling of NK cells operated by viruses is a very actual topic that offers many insights into the etiopathogenesis and therapy of cancer.
Author Response
I thank this reviewer for his/her constructive comments which I have discussed below.
Reviewer #3
In the present paper the author reviews how three different g-Herpesviruses determines a control of the maturation and differentiation of NK cells. In my opinion, the review captures the point proposed in the title and provides a very clear picture of how Herpesviruses are able to modulate the immune response of NK cells.
The strength of this work consists in highlighting how the shaping of NK cells is dependent on the replication phase of Herpesviruses, which until a few years ago was considered a sporadic event and of little interest compared to the latency phase.
The weakness of the work is probably related to the absence of additional figures that could help the reader to promptly acquire information.
I also believe that the remodeling of NK cells operated by viruses is a very actual topic that offers many insights into the etiopathogenesis and therapy of cancer.
I tried to capture the main message of the review in figure 1, namely the expansion of different NK cell subsets during closely related herpesvirus infections. This figure contains indeed new information on CD56-CD16+ NK cell accumulation during KSHV infection that my lab has recently produced (Caduff et al., Cell Rep 2021). I feel that other figures such as for the EBV life cycle and regarding molecules that have been identified in primary immunodeficiencies to predispose for EBV and KSHV pathologies have been recently published by others and myself (Münz, Nat Rev Microbiol 2019; Damania and Münz, FEMS Microbiol Rev 2019). Therefore, these would be redundant and not unique to this review. Thus, I have not included any additional figures in the revised manuscript.